# Experimental design for MRI by greedy policy search

**Tim Bakker,**
University of Amsterdam,
`t.b.bakker@uva.nl`

**Herke van Hoof,**
University of Amsterdam
`h.c.vanhoof@uva.nl`

**Max Welling,**
University of Amsterdam, CIFAR
`m.welling@uva.nl`

## Abstract

In today's clinical practice, magnetic resonance imaging (MRI) is routinely accelerated through subsampling of the associated Fourier domain. Currently, the construction of these subsampling strategies - known as experimental design - relies primarily on heuristics. We propose to learn experimental design strategies for accelerated MRI with policy gradient methods. Unexpectedly, our experiments show that a simple greedy approximation of the objective leads to solutions nearly on-par with the more general non-greedy approach. We offer a partial explanation for this phenomenon rooted in greater variance in the non-greedy objective's gradient estimates, and experimentally verify that this variance hampers non-greedy models in adapting their policies to individual MR images. We empirically show that this adaptivity is key to improving subsampling designs.

## 1 Introduction

Magnetic resonance imaging (MRI) is a non-invasive medical imaging technique with a wide range of diagnostic applications. However, long acquisition times during the imaging process limit patient comfort, throughput, and imaging quality (for instance due to patient movement) [47]. Reducing imaging times has been an active field of research for the past fifty years. A potential avenue for tackling this problem is to acquire less measurement data during a scan, linearly reducing acquisition times: this is often referred to as accelerated MRI [47].

Measurements in MR imaging are performed in the frequency domain, also known as *k-space*. These measurements are transformed (reconstructed) into the familiar MR images through the inverse Fourier transform. Accelerating MRI - reducing data acquisition - amounts to subsampling the k-space, which due to the Nyquist-Shannon sampling theorem will introduce aliasing artefacts in naive reconstructions. The presence of such artefacts renders the resulting images unusable for diagnostic purposes [47]: in order to improve image quality, additional information must be included in the reconstruction process. In clinical settings today, this is typically done using compressed sensing (CS) techniques [26, 6, 8]. With the rise of deep learning (DL), some successes have also been seen using deep reconstruction networks to obtain diagnostic quality images from more aggressively subsampled k-space [18, 25, 16, 34, 14, 43, 30, 29, 40]. The additional information utilised is implicitly learned from training data. Such neural networks are trained by applying predetermined subsampling masks to k-space: from this masked frequency domain the model then learns to reconstruct target images obtained from the fully sampled k-space [47]. In these CS and DL settings, subsampling masks are typically determined beforehand: either carefully crafted by experts, or based on heuristics [19].

A natural next step is a move away from handcrafted subsampling masks towards learned acquisition strategies. The process of choosing an optimal set of measurements is known as experimental design [37]. While such design methods can be employed to learn a fixed subsampling mask for a data set [15, 2, 45, 11, 12, 32], an ostensively more salient approach is to learn an adaptive strategy that has the ability to propose different masks for different MR images (e.g. various patients or locations). Intuitively, such methods should outperform their non-adaptive counterparts on reconstruction quality given the same measurement budget. In the DL literature two trends are noticeable. The first learns

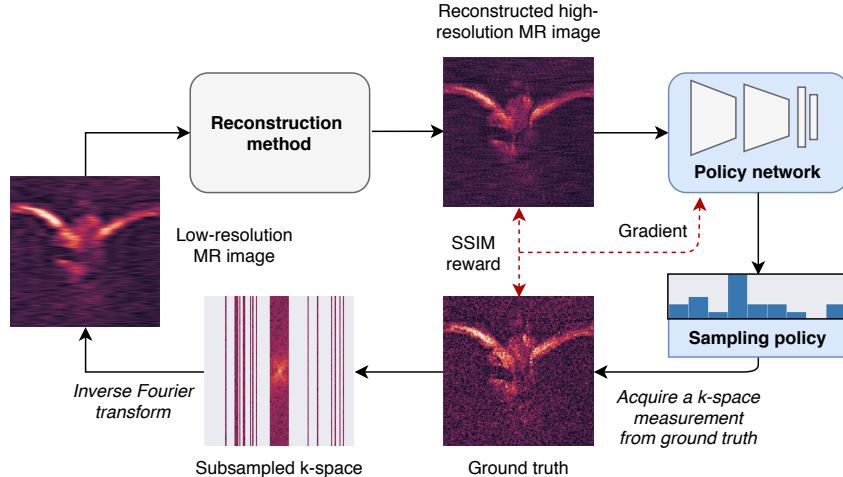

Figure 1: The iterative acquisition procedure. An initial subsampling of k-space is obtained from the ground truth image. The subsampled frequency domain is fed into a reconstruction method, which for neural network-based reconstructions typically starts with an inverse Fourier transform, such that the input and output domain match. This intermediate step results in a low-resolution, so-called *zero-filled* reconstruction [18]. The high-resolution reconstructed MR image is input to the policy network, which outputs a discrete probability distribution that represents the suggested sampling policy. An action is sampled from this policy, corresponding to a measurement of k-space. This measurement is simulated from the ground truth MR image, and the procedure is repeated until the acquisition budget is exhausted. The reward of an acquisition step is given by the improvement in SSIM of the ground truth and reconstruction resulting from that acquisition.

to sequentially acquire measurements in k-space until some budget is exhausted. These adaptive acquisition methods are trained separately [28] or jointly [19] with the reconstruction model, based on some heuristic [33], or a combination of these [48]. The second type of approach involves jointly learning an optimal subsampling and reconstruction by parameterising the mask itself [45, 2, 15]. These directly learned masks are typically not adaptive.

Here, we frame finding an optimal subsampling mask as a reinforcement learning (RL) problem. Our approach - like [19] and (concurrently) [28] - employs a separate reconstruction and policy model. The reconstruction model performs reconstructions from k-space measurements suggested by the policy model. The policy model suggests which measurement to make, based on the current reconstruction. The reward of each of these measure-reconstruct cycles is given as the improvement in reconstruction quality provided by some appropriate metric. The reinforcement learning perspective additionally allows for an analysis of a second split in the literature: greedy and non-greedy models. When optimising an MRI subsampling mask - given some budget - a natural question to ask is whether direct optimisation of the long-term reconstruction quality enjoys strong advantages over a greedy optimisation that acquires the measurement that leads to the greatest immediate improvement.

In the literature, proposed models are often compared to compressed sensing baselines, but rarely to other DL methods due to an historical lack of standardised datasets and evaluation metrics. This limits MRI experimental design research, as it is unclear how different families of methods (e.g. adaptive/non-adaptive, greedy/non-greedy) compare, and thus which directions are the most promising future research targets. We attempt to bridge some of this gap here, in the hope to guide future research into experimental design for MRI. In this paper, we provide a proof of concept showing that direct policy gradient methods can be used to learn experimental design for MRI. We construct both greedy and a non-greedy models using the above-mentioned RL framework. The greedy model is quicker to train and outperforms some of its non-greedy counterparts in our experiments. We hypothesise that the underperformance of non-greedy models is partially due to weaker adaptivity to individual MR slices as a consequence of greater variance in the gradient estimates, and we provide experimental evidence for this hypothesis.[1]

## 2 Related work

With the introduction of compressed sensing, several methods for designing MRI undersampling masks emerged. The original theoretical justification for CS prescribes the use of uniformly random undersampling masks, incidentally removing the need for specific designs [6, 8]. However, most practical success was achieved using variable-density sampling (VDS) heuristics [26], which employ higher sampling density in the low-frequency regions of the Fourier domain. We refer to [17] for an overview. There has also been interest in data-driven mask design for compressed sensing [13, 38, 42]. Such methods often design fixed (non-adaptive) masks that perform well on average over a particular data set. Some recent models in this area include [11, 12, 32], which aim to select an optimal stochastic greedy mask for a fixed budget, using training data as guidance. These models can be applied on top of any reconstruction method, exploiting the recent surge of interest in developing strong (even pre-trained) deep reconstruction models for arbitrary mask designs [18, 25, 16, 34, 14, 43, 30, 29, 40]. The work by [27, 36] instead proposed to use the criterion of maximising posterior information gain to guide a greedy, adaptive mask design.

While deep learning research for undersampled MRI reconstruction has primarily focused on improving reconstruction methods, there has been some success learning undersampling masks in recent years as well. The existing methods can roughly be characterised by three families: non-greedy and non-adaptive, greedy and adaptive, or non-greedy and adaptive.

Notable non-greedy, non-adaptive strategies have been proposed by [15, 45, 2]. Introducing various continuous relaxations of the binary subsampling mask, these works jointly optimise their mask design with a reconstruction network. A greedy, adaptive method by [33] estimates a posterior over undersampled MR reconstructions with a conditional Wasserstein GAN [1], and uses empirical pixel-wise variance of the Fourier domain under this posterior to guide an acquisition strategy. In a similar vein, [48] learns a reconstruction network as well as a pixel-wise uncertainty measure according to the model proposed in [20]. An evaluator network is adversarially trained alongside to learn to recognise mistakes in k-space (implicit in the reconstruction) made by the reconstruction network. The evaluator is used to guide the acquisition process, while the uncertainty measure is used for monitoring only. Joint training of the evaluator network with the reconstruction network is crucial, as the reconstruction network must be incentivised to produce reconstructions that have consistent k-space representation for evaluator based acquisition to perform well.[2]

Unique in the literature is the non-greedy, adaptive method by [19]. Inspired by DeepMind's AlphaZero [39], the authors propose to jointly train a reconstruction and policy network, using Monte Carlo tree search to generate self-supervised targets for the sampling policy. Similar to our proposed method, the reconstruction and sampling models can be decoupled, and as such a sampling strategy can be learned for any given reconstruction method. Unlike our approach however, MCTS based training does not naturally allow for training greedy models. Additionally, our approach enjoys a computational advantage - due to the use of smaller models - and converges more quickly. Finally, direct policy optimisation involves fewer design choices than computing an MCTS distribution.

## 3 Background

### 3.1 Policy gradients and reinforcement learning

We formalise the sequential selection of subsampling masks for MRI as a Partially Observable Markov Decision Process (POMDP). The latent state $z_t$ at acquisition step $t$ of this POMDP corresponds to a tuple $(\boldsymbol{x}, h_t)$ of the true underlying MR image $\boldsymbol{x}$ and the history $h_t$ of actions $a$ and observations $o$: $h_t = (a_0, o_0, ..., a_{t-1}, o_{t-1})$. An action $a$ represents a particular k-space measurement, and the corresponding observation $o$ is the result of that measurement. The agent takes as internal state a summary of the history $h_t$, provided by the current reconstruction $\hat{\boldsymbol{x}}_t$, and outputs a policy $\pi(a_t|\hat{\boldsymbol{x}}_t)$ over actions. The reward $r(z_t, a_t)$ is the improvement in reconstruction quality due to measurement $a_t$ taken when the latent state is $z_t$. The goal here is to learn a policy that maximises the expected sum of rewards (i.e. return) given some measurement budget.

Policy gradient methods are an approach for directly maximising the expected return $J(\phi)$ of a policy $\pi_\phi$ parameterised by $\phi$ on such a POMDP. The log-ratio trick can be used to rewrite the gradient of $J$

with respect to the policy parameters $\phi$ as an expectation of a gradient [41, 3]:

$$\nabla_\phi J(\phi) = \mathbb{E}_{\hat{\boldsymbol{x}}_0} \sum_{t=0}^{T-1} \left[ \nabla_\phi \log \pi_\phi(a_t|\hat{\boldsymbol{x}}_t) \sum_{t'=t}^{T-1} \gamma^{t'-t} \left( r(z_{t'}, a_{t'}) - b(z_{t'}) \right) \right]. \quad (1)$$

Here, $\mathbb{E}_{\hat{\boldsymbol{x}}_0}$ is an expectation over initial states $\hat{\boldsymbol{x}}_0$, and $t \in [0, T-1]$ indexes the time step of an episode. The introduction of a reward baseline $b(z_{t'})$ reduces the typically high variance of this estimator [41, 46]. This baseline can be any function that is independent of the choice of action $a_{t'}$. In our setting, we will construct $b(z_{t'})$ out of rewards obtained from multiple rollouts that start in the same state $z_{t'}$.

Although we are interested in the undiscounted objective ($\gamma = 1$), a discount factor $\gamma < 1$ can help obtain better results by further reducing variance at the cost of a small bias [35, 3]. We are particularly interested in the completely greedy setting - where $\gamma = 0$ - as this leads to particularly simple implementations.

## 3.2 MRI subsampling

We consider a dataset $\mathcal{D}$ of image vectors $\boldsymbol{x} \in \mathbb{R}^N$ corresponding to true MR images to be reconstructed from a subsampled k-space signal. The full k-space signal $\boldsymbol{y}_N \in \mathbb{C}^N$ is obtained from $\boldsymbol{x}$ by a Fourier transform $F \in \mathbb{C}^{N \times N}$ as $\boldsymbol{y}_N = F\boldsymbol{x}$. The subsampling operator (or mask) $U_m$ that selects $m \le M < N$ measurements for a total sampling budget $M$ can be represented as an $m \times N$ matrix with every row a one-hot vector of size $N$. Subsampled k-space can then be written $\boldsymbol{y}_m = U_m F\boldsymbol{x}$. The ratio $N/m$ is called the acceleration of the MR reconstruction process.

We consider subsampling strategies $\pi(\boldsymbol{y}_m)$ that select (possibly stochastically) which k-space measurement should be selected next given the subsampled k-space so far. We would like to find the strategy $\pi$ that yields samples that allow for good reconstructions under reconstruction process $G_\theta$ (where $\theta$ are any parameters in the reconstruction process). The quality of the reconstruction is measured using a quality metric $\eta(\boldsymbol{x}, \cdot)$. Formally, we would thus like to find the strategy that maximises the expected quality after $M$ steps:

$$\pi^* = \arg\max_\pi \eta(\boldsymbol{x}, G_\theta(U_{m+1} F\boldsymbol{x})), \quad U_{m+1} = \begin{bmatrix} U_m \\ k^\mathsf{T} \end{bmatrix}, \quad k \sim \pi(\boldsymbol{y}_m). \quad (2)$$

In this equation, $k$ is a one-hot vector sampled from the multinomial distribution specified by $\pi$. It is possible to optimise the subsampling and reconstruction method jointly, in which case optimisation is over both $\pi$ and $\theta$.

While in principle one may obtain a single element (pixel) of the full 2-dimensional k-space $\boldsymbol{y}_N$ (corresponding to a single MR image) per measurement, it is more common to obtain a full column [47]. Such Cartesian trajectories - consisting of subsequently measured columns - are typically more efficient due to physical constraints in MR machines. Furthermore, the reduction in their scan time is linear in the number of measurements done. Importantly, implementing optimised Cartesian sequences in practice requires only small modifications to MR software [26]. As such, in this work we only concern ourselves with Cartesian samplings, and in what follows have redefined $m, M, N$ as referring to the number of columns in the 2-dimensional k-space.

## 4 Method

We now connect equations (1) and (2). In our approach, the subsampling operator $U_M$ is iteratively constructed by sampling from a policy $\pi_\phi(a_t|\hat{\boldsymbol{x}}_t)$, which is a conditional probability distribution over discrete actions (measurements). This policy is output by a policy network - parameterised by $\phi$ - that takes as input the current internal state $\hat{\boldsymbol{x}}_t$ given by the current reconstruction $G_\theta(U_{t+L} F\boldsymbol{x})$. Here $L$ is the initial number of measurements, such that $\hat{\boldsymbol{x}}_0$ corresponds to $G_\theta(U_L F\boldsymbol{x})$. We will in the following refer to $T = M - L$ - the number of measurements to acquire - as the acquisition horizon.

Starting from an initial subsampling $U_L$, we compute the policy and sample an action $a_0$. This action corresponds to doing a measurement, which we may write as a one-hot vector $k_0$ of size $N$. The subsampling mask for the next acquisition step is then constructed by concatenating this vector in the column direction to the matrix representation of $U_L$ as $U_{L+1} = \begin{bmatrix} U_L^\mathsf{T} k_0 \end{bmatrix}^\mathsf{T}$. This operator is

applied to the ground truth k-space, and a new reconstruction $\hat{\boldsymbol{x}}_1 = G_\theta(U_{L+1}F\boldsymbol{x})$ is obtained. A reward is computed using the criterion $\eta$ and the ground truth image $\boldsymbol{x}$. The reconstruction is again input to the policy network, from which a policy $\pi_\phi(a_1|\hat{\boldsymbol{x}}_1)$ is obtained. This process is repeated until a total of $M$ measurements have been made (including the initial $L$), corresponding to the $T$ steps of equation (1). Equation (2) may now be rewritten as an iterative optimisation over the policy network parameters $\phi$. In the following we use $G_t$ as shorthand for $G_\theta\left(\begin{bmatrix} U_{t-1} \\ k_{t-L-1}^\mathsf{T} \end{bmatrix}F\boldsymbol{x}\right)$ with $G_L = G_\theta\left(U_L F\boldsymbol{x}\right)$, and write $\pi_\phi(G_t)$ for the policy obtained for acquisition step $t$ on MR image $\boldsymbol{x}$:

$$\hat{\phi} = \arg\max_\phi \left\{ \mathbb{E}_{\boldsymbol{x}\sim\mathcal{D}} \sum_{t=L}^{M-1} \pi_\phi(G_t)\left[\eta(\boldsymbol{x}, G_{t+1}) - \eta(\boldsymbol{x}, G_t)\right] \right\} = \arg\max_\phi J(\phi). \quad (3)$$

Here the expected return $J(\phi)$ has been decomposed into separate reward signals over the sequential acquisition steps, by in each step only considering the reconstruction improvement $r^{\boldsymbol{x}}(G_t, G_{t+1}) = \eta(\boldsymbol{x}, G_{t+1}) - \eta(\boldsymbol{x}, G_t)$. The total return is thus given as the total improvement in reconstruction quality over the full acquisition horizon. The MRI subsampling problem has now been formulated as a maximisation of the expected return under a policy given some initial state, and thus we may use gradient ascent on equation (1) to optimise it. For the criterion $\eta$ we use the Structural Similarity Index Measure (SSIM) [44]. The SSIM is a differentiable metric that typically corresponds to human evaluations of image quality more closely than alternatives such as Peak Signal-to-Noise Ratio (PSNR) or Mean-Squared Error (MSE) [22]. Figure 1 summarises the iterative acquisition procedure.

Typically in RL settings a single action is sampled in each time step. However, when multiple actions can be taken in parallel and the rewards for all these actions can be observed, this counterfactual information can be leveraged to construct strong local baselines [23, 24]. As we have access to the ground truth k-space for all our MR images $\boldsymbol{x}$, we may simulate doing multiple acquisitions in parallel, and compute reconstruction improvements for all of them. Sampling $q$ actions at every time step, and writing $r_{i,t}^{\boldsymbol{x}}$ for the reward obtained from sample $i$ at time step $t$, we obtain the following estimators:

$$\nabla_\phi J(\phi) \approx \frac{1}{q-1}\mathbb{E}_{\boldsymbol{x}\sim\mathcal{D}} \sum_{i=1}^{q} \sum_{t=L}^{M-1} \left[ \nabla_\phi \log \pi_\phi(G_t) \sum_{t'=t}^{M-1} \gamma^{t'-t}\left(r_{i,t'} - \frac{1}{q}\sum_{j=1}^{q} r_{j,t'}\right) \right] \text{ (Non-greedy)}, \tag{4}$$

$$\nabla_\phi J(\phi) \approx \frac{1}{q-1}\mathbb{E}_{\boldsymbol{x}\sim\mathcal{D}} \sum_{i=1}^{q} \sum_{t=L}^{M-1} \left[ \nabla_\phi \log \pi_\phi(G_t) \left(r_{i,t} - \frac{1}{q}\sum_{j=1}^{q} r_{j,t}\right) \right] \text{ (Greedy)}. \tag{5}$$

The process for obtaining parallel samples varies between the greedy an non-greedy settings. In the greedy setting, one may easily obtain parallel reward samples by at every time step acquiring $q$ k-space columns in parallel, obtaining the corresponding $q$ reconstructions, and computing the resulting rewards. Since the objective (5) relies only on immediate rewards, gradients can be immediately computed. In the non-greedy setting (4) however, entire trajectories must be sampled before a single optimisation step may be performed. Sampling multiple actions in every state along the trajectory leads to a combinatorial explosion in number of rewards that need to be computed and gradients that need to be stored. To circumvent this issue, in the non-greedy setting we sample multiple actions in the initial state only. This initialises parallel trajectories, which may be traversed using a single sample at every time step (as in more typical reinforcement learning settings). Local baselines for a given time step and trajectory are then computed using the reward corresponding to that time step in the parallel trajectories. Figure 2 illustrates the parallel sampling procedure and associated gradient computation.

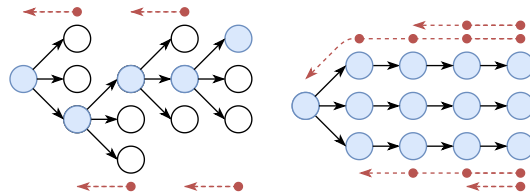

Figure 2: Parallel sampling procedure for the greedy (left) and non-greedy (right) estimators, for an acquisition horizon of four with $q = 3$ samples. Open circles represent states, with blue open circles the states selected to continue a trajectory. Solid black arrows represent actions, dashed red arrows represent gradients flowing from rewards in future states of the acquisition horizon to earlier states. Solid red circles on the gradient arrows denote future rewards included in the gradient.

# 5 Experiments

## 5.1 Implementation

**Datasets:** We leverage the NYU fastMRI open database containing a large number of knee and brain volumes for our experiments [47], restricting ourselves to single-coil scans. The provided test volumes do not contain full ground-truth k-space measurements, so we split 20% of the training data into a test set. For the knee data, we use half of the available volumes for computational expedience, and additionally remove the outer slices of each volume as in [15], since those typically depict background. This leads to a dataset of 6959 train slices, 1779 validation slices, and 1715 test slices. The slices were cropped to the central $128 \times 128$ region as in [19], again to save on computation. We will refer to this dataset as the Knee dataset from now on.

We also construct a Brain dataset from the fastMRI brain volumes. Here the full singlecoil k-space is simulated from the ground truth images by Fourier transform after cropping to the central $256 \times 256$ region. This allows us to test whether policy gradient models can scale to images larger than $128 \times 128$. We use a fifth of the available volumes for computational reasons, resulting in 11312 train slices, 4372 validation slices, and 2832 test slices.

**Reconstruction model:** Jointly training our policy networks with a reconstruction model - while possible - adds variance to the training process: this has effects on the policy models that are hard to predict. Since we are primarily interested in analysing the subsampling strategy, we instead opt to use a pretrained reconstruction model in our experiments. Many powerful reconstruction networks are already available [22], motivating the need for subsampling methods that may be trained on top of any of these models. Additionally, the use of a fixed, pretrained reconstruction model allows us to show that adaptivity indeed improves performance of subsampling methods, by comparing to a non-adaptive oracle that requires a predefined reconstruction method to make predictions. This intuitive notion has, to our knowledge, not yet been empirically quantified. For the reconstruction model we use the standard 16-channel U-Net baseline provided in the fastMRI repository. Hyperparameters and training details are left mostly unchanged: we refer to Appendix A.1 for further details.

**Policy models:** We train greedy and non-greedy policy models using equations (5) and (4) respectively. Setting $\gamma = 1.0$ in equation (4) corresponds to the undiscounted objective, where the policy gradient is an unbiased estimate of the gradient of the expected return: we refer to this as the (fully) non-greedy model. We additionally report results on $\gamma = 0.9$, as this model performed best in our experiments (see Appendix B.5.2 for additional results).

Two initial accelerations are considered: subsampling by a factor 8 and 32. For Knee data, respectively 16 and 28 acquisitions are performed, resulting in a final acceleration of 4 in both cases. We refer to these two settings as respectively the 'base' and 'long' horizon settings. For Brain data we perform the same procedure, but note that here the final mask does not correspond to an acceleration of 4, due to the larger image size. Reconstructions are initialised by obtaining low frequency (center) columns of k-space equal to the initialisation budget. The process as described in section 4 is then performed, with $q = 8$ for both estimators. Models are trained for 50 epochs using a batch size of 16. We use the same architecture for the greedy and non-greedy policy models, as experimentation with larger and smaller architectures showed no clear improvements. Further model details are contained in Appendix A.2 and A.3, as well as a comparison of computational load in B.1. The greedy model is somewhat simpler to implement and significantly less costly to train than the non-greedy model.

**AlphaZero model:** The non-greedy, adaptive, AlphaZero-inspired method of [19] provides a literature baseline for comparison with our models. Their model can be adapted to our task, as it is flexible enough to learn a policy given any reconstruction method and reward signal.[3]

To make a proper comparison, we use the output of our fixed reconstruction model as input to their policy model. The policy model is trained in a self-supervised manner by means of Monte Carlo tree search, using SSIM for the reward signal rather than the original PSNR. In the following, we will refer to this model as AlphaZero. Due to computational constraints, we were only able to train this model on the Knee dataset. Further details can be found in Appendix A.4.

Table 1: SSIM performance on test data. For non-deterministic models, averages and standard deviations are computed over five seeds, using $q = 8$ trajectories for policy models (AlphaZero scores are averaged over three seeds instead).

| | Knee | | Brain | |
|---|---|---|---|---|
| | **Base horizon** | **Long horizon** | **Base horizon** | **Long horizon** |
| **Random** | $0.6948\pm0.0003$ | $0.6602\pm0.0006$ | $0.9020\pm0.0001$ | $0.5820\pm0.0006$ |
| **NA Oracle** | $0.7213$ | $0.7421$ | $0.9099$ | $0.8909$ |
| **AlphaZero** | $0.7203 \pm 0.0008$ | $0.7403 \pm 0.0009$ | - | - |
| **NGreedy** | $0.7223 \pm 0.0003$ | $0.7421 \pm 0.0014$ | $0.9103 \pm 0.0002$ | $0.8886 \pm 0.0048$ |
| **Greedy** | $0.7230 \pm 0.0001$ | $0.7442 \pm 0.0007$ | $0.9106 \pm 0.0001$ | $0.8917 \pm 0.0002$ |
| $\gamma = \mathbf{0.9}$ | $0.7232 \pm 0.0002$ | $0.7449 \pm 0.0004$ | $0.9106 \pm 0.0003$ | $0.8921 \pm 0.0001$ |

## 5.2 Results

In Table 1 we report average and standard deviation test data SSIM scores of the final reconstruction obtained under various models after exhausting the sampling budget. 'Greedy' and 'NGreedy' are our proposed models, and '$\gamma = 0.9$' is the non-greedy model with discount factor $\gamma = 0.9$ instead of $\gamma = 1.0$. 'AlphaZero' refers to the model of [19] adapted to our task as described in section 5.1. Also compared is a non-adaptive oracle, denoted as 'NA Oracle'. This oracle selects as a measurement candidate in each acquisition step the column that leads to the greatest average SSIM improvement over the test dataset. This is not a feasible strategy without direct access to the ground truth k-space, and provides an upper bound on the performance of greedy models that do not adapt their predictions to individual MR slices. Additionally, we compare a 'Random' strategy that shares the initial mask with the other methods, and subsequently obtains a uniformly random measurement every acquisition step, similar to a simple VDS heuristic. See Appendix B.5 for extended results.

Our Greedy model obtains SSIM scores superior to or on-par with most compared models on all tasks. Of the methods reported here, only the $\gamma = 0.9$ model ostensibly outperforms it, although the differences are (close to) within one standard deviation. The Greedy model is furthermore much less computationally expensive to train than any of the non-greedy models. In the following we focus our attention on the Greedy and NGreedy models specifically, as these two extrema prove illustrative for our analysis of adaptivity.

**Adaptivity:** The Greedy model shows in Table 1 that it is adapting its predictions to individual MR images by outperforming the NA Oracle. This suggests that adaptivity is indeed a useful property for subsampling models to possess, and that this information can be learned in practice. Our NGreedy model outperforms the NA Oracle only clearly on the base horizon task, suggesting that for longer horizons it may fail to be sufficiently adaptive. Note that the NGreedy model has two avenues for potentially improving over the NA Oracle: adaptivity, and non-greediness.

To further investigate this behaviour, we visualise the average (over all slices in the Knee dataset) of the learned policies at every acquisition step in Figure 3. Notable is that the NGreedy model outputs more sharply peaked average policies than the Greedy model, suggesting a lack of adaptivity to the input (see Appendix B.2 for more average policy visualisations).

**Mutual information:** As adaptivity is required for a greedy model to outperform the NA Oracle, we may conclude that the Greedy policies are adaptive, but it would be prudent to quantify this effect. The mutual information (MI) can be used as a quantitative measure of how much information an observed state (reconstruction) gives about the action (measurement location) under the learned policy. This mutual information provides a direct measure of how adaptive a model is: the higher the MI, the more the model changes its policy as the state changes. For details see Appendix B.4.1.

In Figure 4 we visualise the MI per acquisition step for the Greedy, NGreedy and $\gamma = 0.9$ models on Knee data. For the base horizon task, it seems the Greedy model learns to be adaptive already in the first acquisition, having enough information to produce adaptive policies. The NGreedy model has low mutual information for the initial acquisition steps, but becomes more adaptive as it gets closer to exhausting the acquisition budget. For the long horizon task the differences in MI are much starker, which is consistent with our observation that the NGreedy model fails to achieve the adaptivity

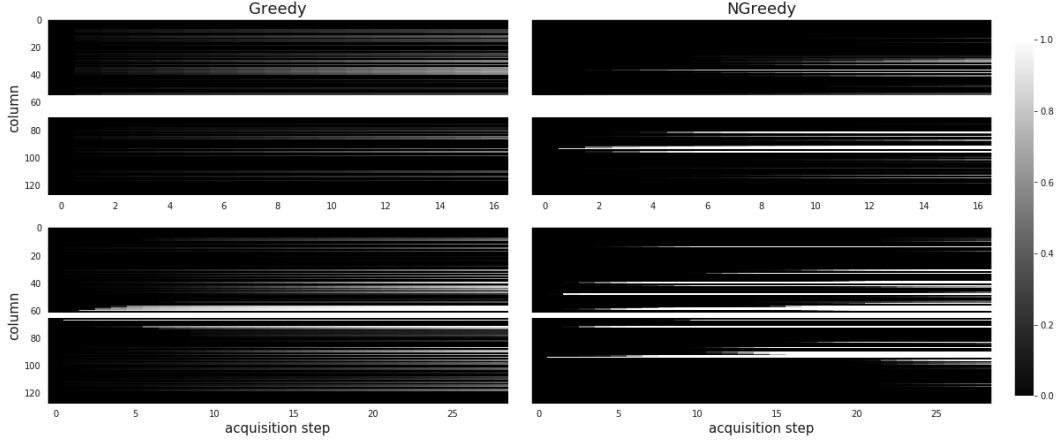

Figure 3: Visualisation of sampled trajectories for base (top) and long (bottom) horizons, averaged over the Knee test data. Shown is the fraction of MR slices for which a particular column has been sampled at an acquisition step. The central white bands are initialisation measurements. The NGreedy policies select the same column for more slices than the Greedy model, suggesting it adapts less to the individual images. For each setting the best model on the test set was used.

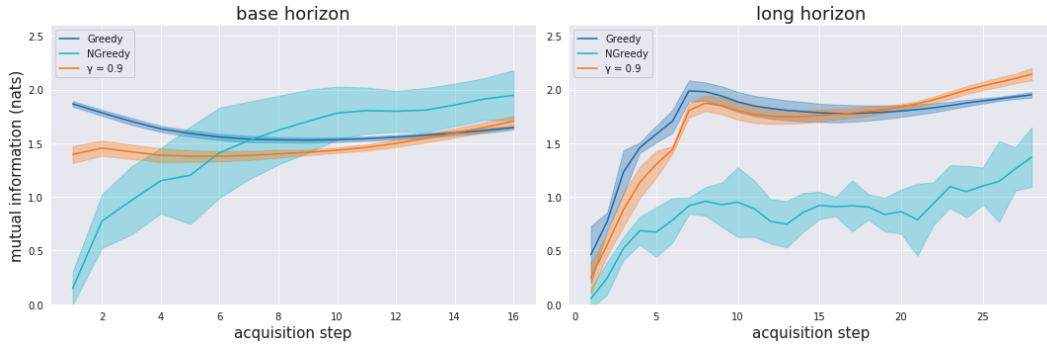

Figure 4: Mutual information for the base (left) and long (right) horizon settings for the Greedy, NGreedy and $\gamma = 0.9$ methods on Knee test data. Shown is the average and standard deviation of the mutual information per acquisition step over five seeds, computed with $q = 8$ trajectories.

required to outperform the NA Oracle. Note that $\gamma = 0.9$ seems to interpolate between Greedy and NGreedy, and behaves more similar to former, even though its objective and training details are more alike the latter. Similar results are obtained for the Brain dataset. We refer to Appendix B.4 for a more comprehensive analysis of the MI.

**Signal-to-Noise:** We hypothesise the lack of adaptivity - of the NGreedy model relative to the Greedy - is partially due to higher variance gradient estimates in the NGreedy estimator, resulting from the longer acquisition horizon. Rather than learning to adapt the policy to individual MR slices, the NGreedy models seems to use the average reward signal to learn policies that perform well on average. The signal-to-noise ratio (SNR) provides a quantitative measure that can be used to compare the variance of our gradient estimators [31]. Taking inspiration from [5], we compute the SNR of a gradient estimate as the empirical mean gradient $\hat{\mu}$ divided by the empirical standard deviation $\hat{\sigma}_\mu$ of this mean gradient. The mean gradient is computed over training data as the average of gradients $g_i$ obtained from batch $i \in [1, B]$: $\hat{\mu} = \frac{1}{B} \sum_{i=1}^{B} g_i$. For large enough $B$, the variance of $\hat{\mu}$ is well estimated as $\hat{\sigma}_\mu^2 = \frac{1}{B(B-1)} \sum_{i=1}^{B} (g_i - \hat{\mu})^2$ due to the Central Limit Theorem [5].

Estimated SNR on the training data is depicted in Table 2. SNR is computed over the gradients of the weights in the final layer of the policy network. The non-greedy models consistently have lower SNR than the Greedy model for both horizons and datasets, consistent with our hypothesis.

Table 2: Signal-to-Noise ratio comparison of the Greedy and NGreedy models for the two time horizons trained on. Displayed are average SNR estimates and standard deviations obtained for the best performing model on three runs over the test data for every setting ($q = 16$ samples, batch size 16). Epoch $n$ refers to the model after the $n$'th epoch of training has been completed.

| | **Knee** | | | | | |
| | **Base horizon** | | | **Long horizon** | | |
| | **Greedy** | **NGreedy** | $\gamma = 0.9$ | **Greedy** | **NGreedy** | $\gamma = 0.9$ |
|---|---|---|---|---|---|---|
| **Epoch 1** | 2.21±0.24 | 1.82±0.01 | 2.22±0.18 | 2.46±0.25 | 1.68±0.14 | 2.05±0.11 |
| **Epoch 20** | 3.91±0.08 | 1.24±0.07 | 2.12±0.06 | 3.49±0.27 | 1.04±0.16 | 3.04±0.06 |
| **Epoch 50** | 2.51±0.03 | 1.02±0.07 | 1.43±0.10 | 2.15±0.16 | 0.96±0.16 | 1.29±0.11 |
| | **Brain** | | | | | |
| | **Base horizon** | | | **Long horizon** | | |
| | **Greedy** | **NGreedy** | $\gamma = 0.9$ | **Greedy** | **NGreedy** | $\gamma = 0.9$ |
| **Epoch 1** | 6.70±0.09 | 3.76±0.22 | 5.31±0.10 | 8.75±0.20 | 1.57±0.11 | 7.80±0.10 |
| **Epoch 20** | 11.21±0.08 | 2.95±0.23 | 5.32±0.02 | 13.36±0.19 | 1.18±0.15 | 7.22±0.19 |
| **Epoch 50** | 7.02±0.07 | 1.45±0.10 | 2.35±0.00 | 4.56±0.09 | 0.82±0.08 | 2.98±0.09 |

Additionally, the lower SNR for NGreedy estimators on the long versus the base horizon task suggests that higher variance is associated with longer horizons, although we note that this effect does not hold for $\gamma = 0.9$ models. This analysis suggests that the non-greedy models obtain lower bias in the optimisation objective at the cost of higher variance, as was already observed in [35]. For these specific experiments, the sweet spot in this trade-off seems to lie around $\gamma = 0.9$ (see Appendices B.5 and B.6).

**AlphaZero:** The AlphaZero method of [19] slightly underperforms our NGreedy model. Although not a policy gradient approach, we expect it experiences some of the same optimisation problems: high variance in the gradients due to optimisation over long acquisition horizons. However, due to computational constraints (these models take up to a week to train) we were not able to do an extensive hyperparameter search, so it may be possible to further improve this model's performance.

## 6    Conclusion and discussion

We have proposed a practical, easy to train greedy model that learns MRI subsampling policies on top of any reconstruction method, using policy gradient methods. In our experiments this greedy model performs nearly on-par with the most performant model tested, is the simplest to implement, and requires fewest computational resources to train. We have furthermore observed that fully non-greedy models perform worse than their greedier counterparts, and hypothesised that this is due to the former primarily learning from an average reward signal, due to high variance in the optimisation. We have provided a number of experiments to support this hypothesis, and have moreover shown adaptivity to individual MR slices to be an useful property in a sampling policy. To our knowledge, this is the first time the MRI subsampling problem has been analysed on these axes in a deep learning context. Our results suggest that future methods for learning subsampling masks in MRI might profit from greedier / discounted optimisation objectives. Additionally, such methods might do well to incorporate adaptivity in their mask designs.

Our analysis was performed on single-coil volumes. We suspect our conclusions will hold in the more clinically relevant multi-coil settings, but this is to be confirmed by future work [47]. The reward baseline used for the greedy model cannot be used for the non-greedy model due to a combinatorial explosion in the required number of sampled trajectories. Future research may investigate incorporating value function learning [41] to estimate the return of a particular node in Figure 2, thus enabling the more local reward baseline to be used for the non-greedy setting as well. Interestingly, for certain types of subsampling problems, theoretical bounds exist that indicate that optimal adaptive greedy strategies perform almost as well as their optimal non-greedy counterparts [10, 9]. Investigating to what degree such bounds hold for the MR subsampling problem is a promising topic for future work as well.

## Broader Impact

Accelerating MRI reconstructions beyond the current standard has the potential to improve patient satisfaction and throughput. In this paper we have attempted to analyse the subproblem of designing optimal sampling strategies with deep learning methods in a way that provides future research with more principled motivations for choosing a particular approach to this problem. However, more work is needed before the insights of this paper become directly clinically relevant. In particular, it is unclear that simple image quality metrics (such as the SSIM used in this work) provide an adequate measure of clinical usability [7]. While the results of the fastMRI reconstruction challenge [22] suggest that the SSIM does provide estimates of image quality consistent with the preferences of radiologists, it is also noted that current reconstruction methods have trouble identifying subtle pathologies, as these are smoothed out by the average signal from the dataset. The authors of [22] suggest that more work is needed relating radiologists' preferences to image quality metrics at the level of diagnostic interpretation. Until then, methods optimised on such metrics should be used in clinical settings only with extreme care, as they risk falsely declaring a patient free of potential health- and life-threatening pathologies.

## Acknowledgements and disclosure of funding

We thank Marco Federici, Shi Hu, Maximilian Ilse, Daniel Worrall, and especially Wouter Kool and Bas Veeling for useful discussions. We are grateful to the Weights&Biases team [4] for providing their experiment tracking software. We would further like to thank the three anonymous NeurIPS2020 reviewers for their detailed and helpful feedback.

This work is supported by the 'Efficient Deep Learning' (EDL, https://efficientdeeplearning.nl) research programme, which is financed by the Dutch Research Council (NWO) domain Applied and Engineering Sciences (TTW).

We furthermore would like to disclose that M. Welling is Vice President of Technologies at Qualcomm Technologies Netherlands, in addition to his university position.

## Footnotes

[1]Code is available at: https://github.com/Timsey/pg_mri.

[2]While we would ideally compare our method to this work, this is infeasible due to reasons discussed in A.6.

[3]We thank the authors for access to their research code and their advice in selecting appropriate hyperparameters. The long horizon setting is inspired by their suggestions.

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
