[Supplementary Material]

# A Implementation details

## A.1 Reconstruction model

As stated in the main text, our reconstruction model is the standard 16-channel U-Net baseline provided in the fastMRI repository pulled in November 2019.[4] Hyperparameters are left unchanged, except for a switch of optimiser from RMSProp to Adam [21], and training for the full 50 epochs rather than doing early stopping based on the validation set.

The input to the model consists of (real-valued) so-called *zero-filled* images, obtained by applying the inverse Fourier transform to subsampled k-space and taking the complex norm of the resulting image. The full k-space is obtained from the ground truth images by Fourier transform after cropping to $(128 \times 128)$ pixels for Knee data, and $(256 \times 256)$ pixels for Brain data. Note that we crop in image space - rather than k-space - which reduces computation while preserving image detail. We train on accelerations $(4, 4, 4, 6, 6, 8)$, with center fractions $(0.25, 0.167, 0.125, 0.167, 0.125, 0.125)$. This means masks contain a low-frequency (central) k-space region that is always sampled, and have up to half of the budget randomly sampled in the remainder of k-space. The random sampling is done so that the reconstruction model learns to reconstruct for a wide variety of masks [47]. In contrast to our policy models, we only have to train the reconstruction model once for each data set, and so we train on all the available training volumes. We use these models pretrained in our further pipeline. The model has 837,635 parameters.

## A.2 Policy model architecture

We use slightly different policy model architecture for the Knee and Brain model, primarily to keep the number of parameters similar. No changes are made to the model architecture when switching between the Base and Long horizon setting.

### A.2.1 Knee model architecture

Starting from the $(128 \times 128)$ reconstructed image, an initial $(1 \times 1)$ convolution is applied to upsample to 16 channels. We follow this by instance normalisation and ReLU activation. We further employ four convolutional blocks, each consisting of a zero-padded $(3 \times 3)$ convolution layer that doubles the number of channels, followed by an instance normalisation, ReLU activation, and $(2 \times 2)$ max-pooling layer.

The resulting $(8 \times 8 \times 256)$ tensor is flattened and fed through a dense layer of 256 neurons, followed by a leaky-ReLU activation with slope 0.01. This is followed by another such layer and activation, before a final dense layer with 128 neurons and a Softmax operation to turn the output into probabilities corresponding to the columns in k-space. The model has 4,685,568 parameters.

As mentioned in the main text, reconstructions are initialised by obtaining low-frequency (center) columns of k-space equal to the initialisation budget. This corresponds to 16 columns for acceleration 8 and to 4 columns for acceleration 32. After initialisation, the process described in section 4 is performed, setting $q = 8$ samples for both estimators. For the size 128 images used in this work, this process corresponds to respectively 16 and 28 acquisition steps, ending at an acceleration factor of 4 for both settings. Models are trained for 50 epochs using a batch size of 16. A single gradient step is performed after accumulating gradients for a full acquisition trajectory.

### A.2.2 Brain model architecture

The differences between the Brain and Knee policy model architectures are slight: the Brain model uses five convolutional blocks of the type described rather than four, and the initial upsampling is to 8 channels, rather than 16. Because the Brain model input is of size the $(256 \times 256)$, these choices ensure the feature representation after the final convolutional block has size $(8 \times 8 \times 256)$ as well. The model has 4,719,552 parameters.

As with the Knee model, we train the Brain model starting with accelerations 8 and 32, acquiring a further 16 and 28 k-space columns with our policy, for the Base and Long horizon cases respectively. Note that since the Brain images are larger, the final state does not correspond to the acceleration

factor 4 of the Knee setting. Instead, we end up with $\frac{256}{8} + 16 = 48$ and $\frac{256}{32} + 28 = 36$ columns for the Base and Long horizon settings respectively, corresponding to acceleration factors of $\frac{256}{48} = 5\frac{1}{3}$ and $\frac{256}{36} = 7\frac{1}{9}$. This choice was made due to computational concern, as training the Brain models takes up to three times as long as training Knee models, and requires more RAM (see section B.1).

### A.3   Policy model hyperparameters

Hyperparameter tuning with random search on Knee data found little performance differences using larger models or longer training. The most influential hyperparameters proved to be learning rate, batch size, and number of samples per acquisition step. Values of the latter two are constrained by memory considerations during training of non-greedy models (see section B.1), and were set to their highest reasonable values of 16 and 8 respectively. Learning rate was further tuned by hand for both Knee models individually. We did not do any additional hyperparameter tuning for the Brain models, opting to use the exact same settings as we used for the Knee models.

The Greedy and NGreedy models are both trained with a learning rate of $5e-5$. The learning rate is decayed once by a factor 10 after 40 epochs for the Greedy model, and decayed a factor 2 every 10 epochs for the NGreedy model, for a total decay rate of 16. Training was done using the Adam optimiser with no weight decay. The $\gamma = 0.9$ model is trained with the same parameter settings as the NGreedy model. Individual test scores of the runs presented in Table 1 of the main text were computed by averaging scores of 8 trajectories.

As it is always known which k-space columns have already been measured, we artificially set the probabilities for these columns to 0 during training and evaluation. This ensures the model is focused on the task of finding the optimal policy, rather than also trying to learn which measurement have already been done given only the reconstructed input image to go on. We have experimented with instead feeding this as extra information to the policy model, but this tended to destabilise training.

### A.4   AlphaZero hyperparameters

The AlphaZero model architecture used is as defined in Figure 6 of [19], with 32 channels rather than the original 64, as we saw no performance differences in initial experiments. It was implemented using the research code provided to us in private communication. Training used the original 540 rounds, but model training was early-stopped when the final SSIM (after the last acquisition step) on the validation set flattened out. Due to computational constraints, we were unable to do a proper hyperparameter search, and as such have left the remaining hyperparameters unchanged from their default values: brief experimentation with slightly changed hyperparameter settings showed no obvious performance differences.

Individual test scores of the runs presented in Table 1 of the main text were computed by averaging scores of 8 trajectories, the same as for the Greedy and NGreedy policy gradient models. The model has 27,558,849 parameters, about five to six times more than our policy models.

### A.5   Policy gradient model pseudocode

In Algorithm 1 we provide pseudocode for a training epoch of our policy gradient models. As illustrated by Figure 2 of the main text, the greedy and non-greedy estimators (5) and (4) compute different reward baselines. In the pseudocode this is controlled by the $IsGreedy$ argument.

**Policy model train epoch**

Algorithm parameters: number of trajectories $q$, discount factor $\gamma$, IsGreedy True/False, number of acquisition steps $T$

Initialise reconstruction model $G$, initial mask $U_0$, train dataset $\mathcal{D}$ containing batches of ground truth MR images $\boldsymbol{x}$, metric $\eta$ (e.g. SSIM)

**foreach** *batch in* $\mathcal{D}$ **do**
      Compute initial reconstructions $g_0 \leftarrow G(U_0 F \boldsymbol{x})$
      Compute initial metric $v_0 \leftarrow \eta(g_0, \boldsymbol{x})$
      **for** $t \in \{1, ..., T\}$ **do**
          Compute policy $\pi(\cdot | g_{t-1})$ given the current reconstruction
          Sample $q$ actions $a_t$ from $\pi(\cdot | g_{t-1})$
          Obtain probabilities $p_t \leftarrow \pi(a_t | g_{t-1})$
          **if** *IsGreedy* **then**
              Append these actions to $q$ copies of $U_{t-1}$ to form $U_t$
          **else**
              **if** $t = 1$ **then**
                  Append these actions to $q$ copies of $U_{t-1}$ to form $U_t$
              **else**
                  Append these $q$ actions to the $q$ instances of $U_{t-1}$ in memory to form $U_t$
              **end if**
          **end if**
          Compute next-step reconstructions $g_t \leftarrow G(U_t F \boldsymbol{x})$
          Compute metrics $v_t \leftarrow \eta(g_t, \boldsymbol{x})$
          Compute rewards $r_t \leftarrow v_t - v_{t-1}$
          Store log probabilities $\log(p_t)$ of actions, and rewards $r_t$
          **if** *IsGreedy* **then**
              Compute loss according to Equation (5)
              Store gradient updates (e.g. loss.backward() in PyTorch)
              Randomly select one of the $q$ copies of $U_t$ to continue with
          **else**
              **if** $t = T$ **then**
                  Compute loss according to Equation (4)
                  Store gradient updates
              **end if**
              Continue with all $q$ instances of $U_t$
          **end if**
          Update policy model weights using stored gradient updates (e.g. optimizer.step() in PyTorch)
      **end for**
**end foreach**

       **Algorithm 1:** Pseudocode for a train epoch of the policy gradient models.

### A.6   Lack of comparison to Zhang *et al.* (2019)

Ideally, we would wish to compare our method to the greedy acquisition method of [48]. Unfortunately, there are a number of reasons that make this infeasible, which we discuss here.

As mentioned in the main text (Section 2), the approach in [48] requires joint training of the reconstruction network with an evaluator network that guides acquisition through a similarity score between ground truth and fantasised k-space. Joint training is crucial, as the reconstruction network must be incentivised to produce reconstructions that have consistent k-space representation for evaluator based acquisition to perform well. This contrasts with our method, where joint training is optional, and our acquisition function is directly (reinforcement) learned using policy gradients on image-space input. This also poses a challenge for making a fair comparison (using the same reconstruction model): the reconstruction model in [48] is incentivised to care about features that are

not necessarily relevant to our policy, and our reconstruction method is not necessarily incentivised to care about features that are crucial to their evaluator.

To explore these differences we performed a proxy comparison using our reconstruction model and replacing their evaluator score with the true spectral map score computed from ground truth images. Using ground truth test images makes this an oracle method - infeasible in practice - but provides an upper bound for the performance of [48] under our reconstruction model, as we now use true spectral map scores, rather than the estimate learned by the evaluator network. However, this oracle method performed far worse than our models, suggesting that the strategy in [48] indeed depends heavily on reconstruction model design choices that force consistency of k-space, as well as on joint training with the evaluator. We also note that there is no code available for [48], further complicating attempts at a fair comparison.

### A.7  Dynamic range and SSIM

SSIM hyperparameters are kept to their original values in [44]. The dynamic range is a dataset dependent hyperparameter of the SSIM metric that encodes the value range of a particular image. For an MRI slice the dynamic range is typically chosen to be maximum pixel value in the corresponding ground truth volume [47].

### A.8  PSNR evaluation

We have used the Structural Similarity Index Measure (SSIM) [44] as a reward signal in this work. The SSIM is a differentiable metric that typically corresponds to human evaluations of image quality more closely than common alternatives [22]. One such alternative is the Peak Signal-to-Noise Ratio (PSNR). While we do not train on this metric, it may still be insightful to evaluate the trained models by it.

Interestingly however, evaluating the SSIM non-adaptive and adaptive oracles on Knee data with PSNR, results in scores of 27.21, 27.50 on the base horizon, and 25.59, 26.13 on the long horizon task respectively. In contrast, SSIM scores were clearly higher for the Knee data long horizon task, and indeed this is what one would expect for oracles. This suggests that SSIM and PSNR care about distinct features (notably, PSNR seems to favour more low-frequency columns), which complicates drawing conclusions from PSNR evaluations of SSIM optimised methods: verification of reconstruction quality by human experts seems necessary in order to draw further conclusions. Because of this complication, we have opted not to report PSNR scores, though note that the provided code does provide PSNR evaluations.

## B  Further analysis

### B.1  Comparison of computational load

As the architecture for the Greedy and NGreedy models is equivalent, differences in computational load required for training the models stem from variations due to the different optimisation objectives. As the Greedy model requires only looking ahead a single acquisition step, we are only required to store the cumulative gradient of the full acquisition trajectory. The NGreedy model however requires storing individual gradient information for all parallel acquisition trajectories and all acquisition steps, as the full return is only known after performing the last measurement of the budget.

These memory requirements constrain the batch size significantly for the NGreedy model - approximately linearly in length of the optimisation horizon - adding to gradient variance. To circumvent this issue and make a proper comparison with the Greedy model, we accumulate gradients for multiple batches, doing an optimisation step only when the effective batch size has reached that of the Greedy model (dividing the accumulated gradients by the number of batches used to accumulate them to properly mimic training over larger batches). However, this procedure effectively increases the number of batches treated sequentially by a factor equal to the ratio in batch size, slowing down training correspondingly.

We note that the memory burden of the NGreedy model may also be heavily reduced by storing (state, action, reward)-transition tuples for every trajectory encountered during an episode, and discarding model gradients at this step. Then, when the episode has concluded, the full return as well as the log

Table 3: Approximate training times in days for policy models on GTX1080Ti GPUs. Number of GPUs used in parallel given in brackets.

| | Base horizon | | Long horizon | |
|---|---|---|---|---|
| | **Greedy** | **NGreedy** | **Greedy** | **NGreedy** |
| **Knee** | 1.0 (1) | 1.5 (1) | 1.5 (1) | 2.0 (2) |
| **Brain** | 2.0 (1) | 3.0 (2) | 4.5 (1) | 5.0 (4) |

Figure 5: Visualisation of sampled trajectories for base (top) and long (bottom) horizons, averaged over the Knee test data. Shown is the fraction of MR slices for which a particular column has been sampled at an acquisition step. The central white bands are initialisation measurements. The $\gamma = 0.9$ average policies interpolate between the Greedy and NGreedy average policies. For each setting the best model on the test set was used.

probabilities of the stored action may be computed and the gradient backpropagated as normal. This requires one more forward pass of the policy model for every action (to compute the log probability gradients), but frees up storage space because these gradients need not be remembered for the full trajectory. Especially for longer horizons this may speed up computation by exploiting the use of larger batches than otherwise possible.

We refer to Table 3 for a comparison of (very) approximate training times on GTX1080Ti GPUs. These numbers are based on observation of the training logs, not on an exact computation, as the latter was rendered impossible due to issues on the GPU clusters that caused training jobs to crash halfway through. The largest source of variance in training time within a setting seems to be related to I/O, likely due to the loading of large MR image files.

## B.2    Additional policy visualisations

Here we present additional policy visualisations that were omitted from the main text. In Figure 5 we compare the Greedy model with the $\gamma = 0.9$ model for Knee data. The latter outputs more sharply peaked average policies than the former, although - as expected - this difference is less stark than that of Figure 3. These visualisations were obtained by running the policy model for all MR slices in the corresponding test dataset and averaging the number of times a particular measurement was performed at a particular acquisition step. Since the acquisitions are themselves policy samples, these visualisations will slightly differ between runs of the same model. Figure 6 shows a comparison of average policies of the Greedy and NGreedy model for the Brain dataset, and Figure 7 shows the Greedy and $\gamma = 0.9$ average policies. We observe a similar contrast here as we did for Knee data, with the NGreedy model outputting more sharply peaked average policies, and the $\gamma = 0.9$ interpolating. Note that the acquisitions done by the models make up a smaller fraction of the initially selected (and total) k-space columns here than in the Knee case, due to the use of larger images.

## B.3    Learning curves

Figure 8 provides training and validation learning curves for the Knee dataset. Note that the Greedy models seem to overfit slightly more than the NGreedy models, consistent with our hypothesis of

Figure 6: Visualisation of sampled trajectories for base (top) and long (bottom) horizons, averaged over the Brain test data. Shown is the fraction of MR slices for which a particular column has been sampled at an acquisition step. The central white bands are initialisation measurements. The NGreedy policies select the same column for various slices more often than the Greedy model, suggesting it adapts less to the individual images. For each setting the best model on the test set was used.

Figure 7: Visualisation of sampled trajectories for base (top) and long (bottom) horizons, averaged over the Brain test data. Shown is the fraction of MR slices for which a particular column has been sampled at an acquisition step. The central white bands are initialisation measurements. The $\gamma = 0.9$ average policies interpolate between the Greedy and NGreedy average policies. For each setting the best model on the test set was used.

higher gradient variance leading to the latter learning from more average reward signals. Figure 9 provides validation learning curves for the Brain dataset. The training data learning curves were not computed for the Brain dataset, as the computation corresponds to a near doubling of the number of training epochs.

## B.4    Policy mutual information

### B.4.1    Mutual information motivation

In section 5.2 we showed the average policy over the test data set for the best Greedy and NGreedy models. This average policy $\pi(a_t)$ for acquisition step $t$ was computed by running a single trajectory for every setting and taking the average over the data set of $N$ points indexed by $i$ as: $\pi(a_t) = \frac{1}{N} \sum_{i=1}^{N} \pi(a_t|s_{i,t})$, where $s_{i,t}$ is the state corresponding to MR image $i$ at acquisition step $t$.

As stated, the Greedy policy seemed to be more adaptive as its average policy was less peaked that that of the NGreedy policy. However, inspection of $\pi(a_t)$ on its own is not sufficient to support this claim, as the uncertainty in the policy could also be explained as the Greedy model having high uncertainty in $\pi(a_t|s_t)$: that is, high uncertainty on which action to sample even given the current reconstruction.

In the main text we further supported our adaptivity claim by comparing the Greedy model to a non-adaptive oracle, which could only be outperformed by being adaptive to the current state, which

Figure 8: Learning curves on the Knee data train and validation set. For all methods the training SSIM increases steadily with the number of epochs trained, but the effect is stronger the greedier the model, suggesting the Greedy model overfits more strongly than the NGreedy model.

Figure 9: Learning curves on the Brain data validation set. The Greedy model converges quickest, with $\gamma = 0.9$ interpolating between Greedy and NGreedy.

in turn implies low uncertainty in $\pi(a_t|s_t)$ relative to $\pi(a_t)$. Here we directly measure the gap between these two uncertainties for the Greedy and NGreedy model in both the base and long horizon settings.

We use the entropy as a quantitative measure of the uncertainty in the policies. The entropy $H(A_t)$ of a probability distribution over possible actions $A_t$ is known as the marginal entropy, computed as $H(A_t) = \sum_j \pi(a_{j,t}) \log \pi(a_{j,t})$, where $j$ indexes the actions in $A_t$. The conditional entropy $H(A_t|S_t)$ of a conditional probability distribution over actions $A_t$ given a state $s_t \in S_t$ is computed as $H(A_t|S_t) = \sum_j \pi(a_{j,t}|s_t)p(s_t) \log \pi(a_{j,t}|s_t) = \frac{1}{N} \sum_{i,j} \pi(a_{j,t}|s_{i,t}) \log \pi(a_{j,t}|s_{i,t})$, again for the $N$ data points in the test set.

The gap between these two entropies is the mutual information (MI) $I(A_t; S_t)$ of $A_t$ and $S_t$: $I(A_t; S_t) = H(A_t) - H(A_t|S_t)$. This is a quantitative measure of how much information the state gives about the action, under the learned policy. This mutual information provides a direct measure of how adaptive a model is: the higher $I(A_t; S_t)$, the more the model changes its policy as the state changes.

Note that high mutual information does not on its own equal strong performance on the MRI subsampling task. As a degenerate example, consider the case of a specific Knee policy that performs a single acquisition step on 128 slices. For every slice this policy suggests a different measurement with probability 1. The marginal entropy of this policy is $\ln(128)$, and the conditional entropy is 0. This gives the maximum possible mutual information of $\ln(128)$ for this setting - and indeed this policy is maximally adaptive - but clearly this is a bad policy for the task at hand. Of course, as our policies are learned based on the SSIM reward signal, they are incentivised to be adaptive only if this helps the MRI task, and thus the MI gives information about the degree to which these policies are usefully adaptive.

### B.4.2 Knee data mutual information

In Figure 4 of the main text we visualise the MI per acquisition step for the Greedy, NGreedy and $\gamma = 0.9$ models on Knee data. For the base horizon task, it seems the Greedy model learns to be

Figure 10: Marginal and conditional entropies for the base (left) and long (right) horizon settings for the Greedy, NGreedy and $\gamma = 0.9$ methods on Knee test data. Shown is the average and standard deviation of the entropies per acquisition step over five seeds, computed with $q = 8$ trajectories.

adaptive already in the first acquisition, having enough information to produce adaptive policies. The NGreedy model has low mutual information for the initial acquisition steps, but becomes more adaptive as it gets closer to exhausting the acquisition budget.

An explanation for the NGreedy model's behaviour may go as follows: if little is known about an MR image, one might reasonably default to taking measurements in the low-frequency bands, until enough information has been acquired to focus on image-specific details. However, as the Greedy model has high mutual information already at acquisition initialisation, it is more likely that the NGreedy model performs better at later acquisition steps due to the shorter time horizon for optimisation. This is consistent with our hypothesis that gradient variance due to long optimisation horizons hampers the NGreedy model. The NGreedy model eventually overtakes the Greedy model in mutual information: it may be that the NGreedy model has the potential to be more adaptive than the Greedy model for certain acquisition steps due to its non-greediness. Another possible explanation is that the usefulness of adaptivity decreases as more measurements have already been adaptively sampled, and as such the NGreedy model catches up to the Greedy model as the latter runs into diminishing marginal returns more quickly.

Interesting is the bowl-like shape in the Greedy model's MI. It implies higher adaptivity at the start and end of the acquisition trajectory, but whether this is due to properties of the problem setting or due to properties of training is unclear and left for future research.

For the long horizon task, the Greedy model's mutual information steeply climbs during the first few acquisitions, suggesting that initially it does not have enough information to properly adapt its policies to the input: this behaviour mirrors that of the NGreedy model in the base horizon case. The low mutual information means the marginal and conditional entropies are close together in value. To further analyse this behaviour, we show the conditional and mutual entropies separately in Figure 10. Since all entropies are relatively low at acquisition initialisation of the long horizon task, we conclude both models start out selecting a small set of similar measurements for most MR images initially, corresponding to more sharply peaked conditional policies. In the base horizon setting the Greedy model starts out with relatively high values for both entropies, suggesting more adaptivity for this setting. Figure 10 also shows that the NGreedy model generally seems more certain (lower conditional entropy) about its predictions than the Greedy model. This could be related to the NGreedy model learning from an average reward signal, but there are likely other explanations consistent with the current observations as well. While it might also be indicative of the NGreedy model overfitting relative to the Greedy model, we do not observe this here, as shown in Figure 8.

In the long horizon task, the Greedy model obtains the same level of adaptivity as on the base horizon task through the first few acquisitions. After this, it shows the same bowl-like MI shape observed for the base horizon task. The NGreedy model shows similar behaviour as it did on the base horizon task, presenting stronger adaptivity for the final few acquisition steps, likely due to shorter optimisation horizons. Unlike in the base horizon task, it does not catch up to the Greedy model in mutual information for any acquisition step. This provides another indication that the NGreedy model's lack of adaptivity compared to the Greedy model is correlated to longer optimisation horizons.

Figure 11: Mutual information for the base (left) and long (right) horizon settings, for the Greedy, NGreedy and $\gamma = 0.9$ methods on Brain data. Shown is the average and standard deviation of the mutual information per acquisition step over five seeds, computed with $q = 8$ trajectories.

Figure 12: Mutual information for the base (left) and long (right) horizon settings, for the Greedy, NGreedy and $\gamma = 0.9$ methods on Brain data. Shown is the average and standard deviation of the mutual information per acquisition step over five seeds, computed with $q = 8$ trajectories.

The behaviour of the $\gamma = 0.9$ MI and entropies is more similar to that of the Greedy model than that of NGreedy model, indicating that the learned policies exploit similar adaptivity information, while retaining a less Greedy optimisation horizon. This is somewhat surprising, as its optimisation objective and practical training details are much more similar to the NGreedy model than to the Greedy model. Interestingly, the $\gamma = 0.9$ model manages to surpass the Greedy model in mutual information near the end of the acquisition horizon for both settings, which may contribute to its higher performance.

### B.4.3 Brain data mutual information

In Figures 11 and 12 we present the mutual information and entropy plots for the Greedy and NGreedy model trained on Brain data. The behaviour of these quantities is quite similar to their Knee data counterparts in Figures 4 and 10: the Greedy model generally enjoys higher average MI than the NGreedy model, as well as lower variance. The most notable differences are the lack of bowl-like shape in the Greedy MI plots, as well as the inability of the NGreedy MI to overtake the Greedy even in the base horizon setting. Whereas for the Knee dataset the base horizon setting corresponds to a final acceleration factor of 4, for the Brain dataset the final acceleration factor is $\frac{256}{32+16} = 5\frac{1}{3}$, so this comparison is not strictly fair. We note however than for the Knee dataset base horizon setting, the NGreedy model MI overtakes that of the Greedy model already after only half the total acquisition steps have been performed, corresponding to the same acceleration factor $\frac{128}{16+8} = 5\frac{1}{3}$. Indeed, looking back at Table 1, the relative gap in performance between the Greedy and NGreedy model is larger for the Brain dataset than for the Knee dataset.

In the long horizon setting, the MI of the Greedy model requires about twice the number of acquisition steps (compared to Knee data) to reach the point where it flattens out. Since the Brain images contains twice as many columns, this corresponds to the same relative acceleration (around $\frac{1}{12}$).

Figure 13: Mutual information for the base (left) and long (right) horizon settings for the NGreedy method with various values of the discount factor $\gamma \in [0, 0.5]$ on Knee test data. Shown is the average and standard deviation of the mutual information per acquisition step over five seeds, computed with $q = 8$ trajectories.

Figure 14: Marginal and conditional entropies for the base (left) and long (right) horizon settings for the NGreedy method with various values of the discount factor $\gamma \in [0, 0.5]$ on Knee test data. Shown is the average and standard deviation of the mutual information per acquisition step over five seeds, computed with $q = 8$ trajectories.

The $\gamma = 0.9$ Brain models behave broadly similarly to the Knee case, with the exception of the heavy drop and recovery of the marginal entropy late in the long horizon setting. We leave exploration of this behaviour to future work. We furthermore note that in the long horizon setting the $\gamma = 0.9$ MI starts out higher than the Greedy MI. It seems the longer optimisation horizon does not hamper the adaptivity of this model as much, perhaps due to the lower acceleration factor relative to the Knee setting.

### B.4.4 Mutual information on Knee data for various discount factors

In Figures 13, 15, 14, and 16, we present mutual information, conditional entropy, and marginal entropy plots for NGreedy models trained on Knee data with various values of the discount factor $\gamma$. These models represent an interpolation between the Greedy and NGreedy model. See section B.5.2 for SSIM performance of these models. Note that the $\gamma = 0$ model does not correspond to the Greedy model due to a difference in computation of the reward baseline (see Figure 2).

In Table 5, NGreedy models with $\gamma \in [0, 0.5]$ all perform quite similarly to the Greedy model, and indeed their MI and entropy curves look very similar to the Greedy model's. For $\gamma \in [0.5, 1]$ the MI and entropy curves look more like interpolations between the Greedy and NGreedy case for both horizon settings. However, the behaviour surprisingly still seems more similar to the Greedy model than the NGreedy model even for $\gamma = 0.99$.

Figure 15: Mutual information for the base (left) and long (right) horizon settings for the NGreedy method with various values of the discount factor $\gamma \in [0.5, 1]$ on Knee test data. Shown is the average and standard deviation of the mutual information per acquisition step over five seeds, computed with $q = 8$ trajectories.

Figure 16: Marginal and conditional entropies for the base (left) and long (right) horizon settings for the NGreedy method with various values of the discount factor $\gamma \in [0.5, 1]$ on Knee test data. Shown is the average and standard deviation of the mutual information per acquisition step over five seeds, computed with $q = 8$ trajectories.

## B.5 Additional SSIM results

### B.5.1 Extended SSIM table

We report extended SSIM results in Table 4. Here we include results for a greedy oracle model, that selects the measurement leading to the greatest immediate SSIM improvement for every slice separately. We denote this model as 'Oracle'. In principle this is an upper bound on any greedy model, but note that this method is most susceptible to failing to identify situations where a combination of two measurements that are separately uninformative lead to strong improvements. This seems to be the likely explanation for the low Oracle performance in the long horizon Brain setting. Nevertheless, these scores provide an indication of a performance gap that may still be closed by future research.

Additionally, we include a comparison with equispaced masks, which are generally easier to implement in MRI machines than random masks, and may perform better as noted in [40]. As with the random baseline, we initialise these masks by sampling low frequency bands up to the starting acceleration. The two-sided equispaced mask 'Equi (two)' is then constructed by sampling every $r$'th column of the remaining k-space, where $r$ is determined by dividing the number of initially unsampled columns by the number of acquisitions to be made. The one-sided equispaced mask 'Equi (one)' is constructed similarly, but only considering the remaining columns on one side of k-space. One-sided sampling can be more efficient due to k-space symmetry in some cases, as we observe in Table 1.

Table 4: SSIM performance on test data. For non-deterministic models, averages and standard deviations are computed over five seeds, using $q = 8$ trajectories for policy models (AlphaZero scores are averaged over three seeds instead).

| | Knee | | Brain | |
| --- | --- | --- | --- | --- |
| | **Base horizon** | **Long horizon** | **Base horizon** | **Long horizon** |
| **Random** | 0.6948±0.0003 | 0.6602±0.0006 | 0.9020±0.0001 | 0.5820±0.0006 |
| **Equi (one)** | 0.7049 | 0.6880 | 0.9038 | 0.5862 |
| **Equi (two)** | 0.7064 | 0.6918 | 0.9016 | 0.6026 |
| **NA Oracle** | 0.7213 | 0.7421 | 0.9099 | 0.8909 |
| **Oracle** | 0.7379 | 0.7623 | 0.9141 | 0.8872 |
| **AlphaZero** | $0.7203 \pm 0.0008$ | $0.7403 \pm 0.0009$ | - | - |
| **NGreedy** | $0.7223 \pm 0.0003$ | $0.7421 \pm 0.0014$ | $0.9103 \pm 0.0002$ | $0.8886 \pm 0.0048$ |
| **Greedy** | $0.7230 \pm 0.0001$ | $0.7442 \pm 0.0007$ | $0.9106 \pm 0.0001$ | $0.8917 \pm 0.0002$ |
| $\gamma = 0.9$ | $0.7232 \pm 0.0002$ | $0.7449 \pm 0.0004$ | $0.9106 \pm 0.0003$ | $0.8921 \pm 0.0001$ |

Table 5: Average SSIM performance on Knee test data for NGreedy models trained with various $\gamma \in [0, 1]$, the NGreedy model reported in the main text ($\gamma = 1.0$) and the Greedy model. Averages and standard deviations are computed over five seeds, with $q = 8$ trajectories.

| | Knee | |
| --- | --- | --- |
| | **Base horizon** | **Long horizon** |
| **Greedy** | $0.7230 \pm 0.0001$ | $0.7442 \pm 0.0007$ |
| $\gamma = 0.0$ | $0.7231 \pm 0.0002$ | $0.7447 \pm 0.0002$ |
| $\gamma = 0.1$ | $0.7230 \pm 0.0001$ | $0.7446 \pm 0.0010$ |
| $\gamma = 0.2$ | $0.7230 \pm 0.0001$ | $0.7445 \pm 0.0003$ |
| $\gamma = 0.5$ | $0.7230 \pm 0.0002$ | $0.7443 \pm 0.0006$ |
| $\gamma = 0.8$ | $0.7231 \pm 0.0002$ | $0.7445 \pm 0.0004$ |
| $\gamma = 0.9$ | $0.7232 \pm 0.0002$ | $0.7449 \pm 0.0004$ |
| $\gamma = 0.95$ | $0.7232 \pm 0.0001$ | $0.7446 \pm 0.0002$ |
| $\gamma = 0.99$ | $0.7228 \pm 0.0004$ | $0.7437 \pm 0.0007$ |
| **NGreedy** | $0.7223 \pm 0.0003$ | $0.7421 \pm 0.0014$ |

### B.5.2 Knee data SSIM results for other discount factors

In Table 5 we report average SSIM performance for NGreedy models trained on Knee data with various discount factors $\gamma \in [0, 1]$. These models represent an interpolation between the Greedy and NGreedy model. As noted in the main text, the model with $\gamma = 0.9$ performs best. Note the relatively heavy drop in performance very close to $\gamma = 1.0$, which corresponds to our NGreedy model. Out of all the tested discount factors it seems that a fully NGreedy ($\gamma = 1.0$) model performs worst on the Knee dataset.

Note that the $\gamma = 0$ model does not exactly correspond to the Greedy model due to a difference in computation of the reward baseline (see Figure 2), but is otherwise equivalent. We note here that the performance of this model is slightly higher than that of the Greedy model, though still within one standard deviation. The weaker baseline of the $\gamma = 0$ model, may in fact help the optimisation by escaping local minima, as these Greedy models tend to have high SNR and thus seem to be less hampered by variance than the models with higher values for $\gamma$. The different reward baseline may matter more for the non-greedy models, as they seem more troubled by gradient variance. Valuable future research may be to investigate methods for adapting this baseline to the non-greedy case, for instance by incorporating value function learning to estimate the return of a particular node in Figure 2. Due to computational constraints, we do not report a table like Table 5 for Brain data.

While the Greedy model underperforms most of the non-greedy models, the differences are in most cases within one standard deviation of performance. As the Greedy model is much less computationally intensive, the greedy approach may be favoured for certain MRI tasks.

Table 6: Signal-to-Noise ratio comparison of the Greedy, NGreedy and $\gamma = 0.9$ models on both datasets, for the two time horizons trained on. Displayed are average SNR estimates obtained for the best performing model (on test data) on three runs over the train data for every setting, at various points during its training: Epoch $n$ refers to the model after the $n$'th epoch of training is completed. SNR is estimated using gradients for the final layer of the policy network, with $q = 16$ samples and batch size 16 in all settings.

| | Knee | | | | | |
| | Base horizon | | | Long horizon | | |
| | **Greedy** | **NGreedy** | $\gamma = 0.9$ | **Greedy** | **NGreedy** | $\gamma = 0.9$ |
|---|---|---|---|---|---|---|
| **Epoch 1** | 2.21±0.24 | 1.82±0.01 | 2.22±0.18 | 2.46±0.25 | 1.68±0.14 | 2.05±0.11 |
| **Epoch 10** | 3.20±0.10 | 1.17±0.03 | 2.00±0.22 | 5.90±0.14 | 1.16±0.18 | 2.00±0.05 |
| **Epoch 20** | 3.91±0.08 | 1.24±0.07 | 2.12±0.06 | 3.49±0.27 | 1.04±0.16 | 3.04±0.06 |
| **Epoch 30** | 5.90±0.12 | 1.08±0.20 | 2.20±0.12 | 4.78±0.13 | 1.04±0.20 | 2.24±0.21 |
| **Epoch 40** | 2.80±0.40 | 1.02±0.12 | 1.59±0.03 | 2.14±0.08 | 1.06±0.08 | 1.51±0.12 |
| **Epoch 50** | 2.51±0.03 | 1.02±0.07 | 1.43±0.10 | 2.15±0.16 | 0.96±0.16 | 1.29±0.11 |

| | Brain | | | | | |
| | Base horizon | | | Long horizon | | |
| | **Greedy** | **NGreedy** | $\gamma = 0.9$ | **Greedy** | **NGreedy** | $\gamma = 0.9$ |
|---|---|---|---|---|---|---|
| **Epoch 1** | 6.70±0.09 | 3.76±0.22 | 5.31±0.10 | 8.75±0.20 | 1.57±0.11 | 7.80±0.10 |
| **Epoch 10** | 8.44±0.06 | 2.48±0.11 | 4.91±0.08 | 12.10±0.14 | 1.96±0.07 | 6.28±0.17 |
| **Epoch 20** | 11.21±0.08 | 2.95±0.23 | 5.32±0.02 | 13.36±0.19 | 1.18±0.15 | 7.22±0.19 |
| **Epoch 30** | 14.50±0.09 | 2.02±0.21 | 3.25±0.05 | 16.16±0.18 | 1.15±0.12 | 6.50±0.07 |
| **Epoch 40** | 14.91±0.11 | 1.83±0.14 | 3.08±0.23 | 13.04±0.04 | 1.04±0.14 | 3.73±0.26 |
| **Epoch 50** | 7.02±0.07 | 1.45±0.10 | 2.35±0.00 | 4.56±0.09 | 0.82±0.08 | 2.98±0.09 |

### B.6 More SNR results

We report SNR values as in Table 2 for additional intermediate training stages in Table 6. These results are consistent with the conclusions stated in the main text. Shortly after initialisation all Knee settings have relatively similar SNR, likely due to the average reward signal dominating all settings under the initial random policy. For the Greedy estimators, it is notable that the SNR seems to rise and fall quite sharply. We suspect this effect is related to convergence, and we leave closer investigation of it to future work.

### B.7 Reconstruction examples

Here we present some example reconstructions for the Greedy and NGreedy model on both acquisition horizons. Figures 17 and 18 each show a slice of knee data. Figures 19 and 20 instead each show a slice of brain data.

In every image we present from top to bottom: base horizon Greedy, base horizon NGreedy, long horizon Greedy, long horizon NGreedy. From left to right: final subsampling mask, reconstruction at this point, target image, absolute difference between target and reconstruction. For the Brain data long horizon setting, the policies primarily suggest to sample low frequency bands, as was already observed in Figure 6.

Figure 17: Visualisation of a slice of Knee data for various settings. From top to bottom: base horizon Greedy, base horizon NGreedy, long horizon Greedy, long horizon NGreedy. From left to right: final subsampling mask, reconstruction at this point, target image, absolute difference between target and reconstruction.

Figure 18: Visualisation of a slice of Knee data for various settings. From top to bottom: base horizon Greedy, base horizon NGreedy, long horizon Greedy, long horizon NGreedy. From left to right: final subsampling mask, reconstruction at this point, target image, absolute difference between target and reconstruction.

Figure 19: Visualisation of a slice of Brain data for various settings. From top to bottom: base horizon Greedy, base horizon NGreedy, long horizon Greedy, long horizon NGreedy. From left to right: final subsampling mask, reconstruction at this point, target image, absolute difference between target and reconstruction.

Figure 20: Visualisation of a slice of Brain data for various settings. From top to bottom: base horizon Greedy, base horizon NGreedy, long horizon Greedy, long horizon NGreedy. From left to right: final subsampling mask, reconstruction at this point, target image, absolute difference between target and reconstruction.

## Footnotes

[4]https://github.com/facebookresearch/fastMRI/tree/a55a1b129eb1d98ec9df26bfa2617a3b8c957d21