[Reviews · NeurIPS 2020]

Review 1

Summary and Contributions: This paper proposes a policy gradient approach active sampling for deep-learning accelerated MRI.

Strengths: I like the approach this paper takes. Applying policy gradient methods to sample selection makes a lot of sense, and they show an interesting and possibly unexpected result. The paper uses standard baselines and shows clear experimental results. When the simplifying assumption of a fixed pertained model is made, I feel there are some relations to other problems that could be considered or mentioned. For example, the feature selection problem in machine learning is very similar and has been heavily studied. The are various "subset selection" problems that are related. Although, there is not much mention of these related problems in other papers on active acquisition.

Weaknesses: The use of a single unet model is reasonable given it's not the main focus of the paper, however it would be nice to see a model closer to the state-of-the-art used instead. The reconstructions from the single unet are very far behind the SOTA. Using a fixed pretreated model is obviously not ideal, however training a model concurrently with an adaptive sampling pattern is probably impractical. It would be nice to also include equispaced masks as a baseline in the experiments, they generally out-perform random significantly, as noted in [37]. Offset equispaced masking is often even more effective (https://arxiv.org/pdf/1912.01101.pdf)

Correctness: Looks correct.

Clarity: The paper is relatively clear, although the discussion of policy gradient may be a little verbose given how standard the technique has become

Relation to Prior Work: Discussion and citation of related work is very comprehensive.

Reproducibility: Yes

Additional Feedback: There is a arxiv preprint which appears to be concurrent work to this, which uses RL for the same problem: https://arxiv.org/pdf/2007.10469.pdf. Following NeurIPS policies, this is not considered existing work for the purposes of novelty, although it would be good to cite it in the camera ready. UPDATE: I think this paper is relevant to NeurIPS, and I have discussed this with the other reviewers.


Review 2

Summary and Contributions: This paper presents a reinforcement learning (RL) approach for adaptive MRI experimental design, whereby the choice of frequency to scan is conditioned on current partial reconstructions, on a per-image basis. The paper formulates this problem as a Markov Decision Process (MDP), and derives a policy gradient algorithm for solving it. The experiments study the benefits of adaptive policies by comparing them with a non-adaptive one-step oracle, with the results suggesting that the proposed approach is beneficial. The results also compare favorably to those of a recent adaptive approach.

Strengths: * This paper makes a valuable contribution to the literature on MRI reconstruction and acceleration. In particular, deep learning (DL) reconstruction approaches for MRI have seen success in recent years (illustrated by results of the NeurIPS 2019 fastMRI challenge), but the study of deep adaptive MRI acquisition methods is still in its infancy. Such adaptive policies have implications to improve personalized healthcare and reduce patient discomfort. However, to date there are only few pieces of work tackling this topic in the DL literature, and to the best of my knowledge, no published work doing it from an RL perspective. As such, this paper contribution is both significant and novel. It is also of relevance to the NeurIPS community, evidenced by the fastMRI NeurIPS challenge mentioned above (using the same dataset as this paper), and an upcoming version for 2020 using brain data. * The proposed reinforcement learning method is sensible and the empirical results validate its potential. The experimental section is reasonably compelling, as they compare favorably with one of the only two recent DL-based adaptive MRI acquisition methods that I'm aware of. Furthermore, comparison with a greedy non-adaptive oracle helps bring forward the point that adaptive policies are useful and worth exploring in more depth (even beyond their proposed method). Additionally, their results suggesting that greedy policies outperform non-greedy policies is interesting, and I appreciate the experimental analysis done to investigate this point further.

Weaknesses: * In the related work section, the comparison between the present approach and the two most similar approaches [18, 44] is lackluster. In the case of [44] this is particularly important, since there was no empirical comparison to that work. There is also no explanation as to why such a comparison was not done. * The experimental results are done on only one MRI dataset, and images are cropped to a small size (128x128) for computational reasons (raw k-space data in fastMRI dataset is of size 368x640). Therefore, the experiments are done on a highly simplified setting, and it is hard to infer how well these results would scale to a more practical setting. * The differences between the AlphaZero approach and the proposed method are statistically significant but the effect is small, particularly considering the gap with respect to the adaptive greedy method. * The analysis of greedy vs. non-greedy is interesting, but the conclusions that can be drawn from it are still limited. It would have been valuable to study whether properties of the data favor the use of greedy policies. It is possible that the higher variance of the non-greedy method offers only a partial explanation for this result.

Correctness: There are some subtleties hidden in the MDP formulation. I'd argue that the problem, as formulated, is non-Markovian, since the reward function depends on the ground truth image, which cannot be accessed from state information alone. Formally, the function r(s, s') could return more than one value for a given s, s' pair, in the case where two or more ground truth images were consistent with the observations in s'. The text also mentions that "...an action a can lead to various next states s' and rewards", but then this point seems to be ignored in the policy gradient formulation, which, as far as I can tell, implicitly assumes that the transition function p(s'|s,a) is deterministic. That being said, I think these are technical details that would not ultimately affect much in the algorithmic implementation, but they do detract from the presentation in the paper and should be corrected.

Clarity: The paper is for the most part well written, although some aspects can be improved: - According to the text preceding it, Eq. (2) is supposed to define the policy as the one that maximizes quality after M steps, yet the equation uses lowercase m. This is confusing since the first paragraph in Section 3.2. specifically defines m \leq M, where M is the budget. Is the "m" in Eq. (2) a typo? If not, is there missing an additional maximization over all possible values of m from 1 to M? If the equation is correct, then this point can probably be explained more clearly. - Is there a \gamma missing from Eq. (5)? If not, then there is some explanation missing as to why this is not necessary for the non-greedy policy gradient formulation.

Relation to Prior Work: I'm not an expert in the vast MRI acquisition literature, but the discussion of related work appears thorough, including positioning of this work with respect to pre-DL work on mask design with compressed sensing using both non-adaptive and adaptive methods. The discussion also includes several non-adaptive DL-based acquisition methods, and also a discussion of the closest approaches to the paper in the DL literature [18, 44]. However, as mentioned above, the differences of the proposed method and [18, 44] should be discussed in more detail.

Reproducibility: Yes

Additional Feedback: - Is the use of a random policy just a by product of a policy gradient formulation? Is there a reason why a deterministic policy wouldn't be better? (Note that there are also deterministic policy gradient methods available, such as DDPG). - Which specific dataset from the fastMRI data was used? Was it DICOM knee data or was it raw k-space data? This is important, since, as mentioned above, raw k-space data is significantly larger than 128x128. If you are using raw data, can you discuss how your method scales to larger sizes images? ===== POST-REBUTTAL ====== Thanks to the authors for their careful rebuttal. I will keep the score provided in my review, recommending acceptance, and, in general, I'm very positive about this submission and I hope to see it published at NeurIPS. Nevertheless, I agree with other reviewers about the claims regarding greedy vs. non-greedy being insufficiently supported. For example, the statement "In this paper, we show that greedy methods outperform non-greedy methods on the MRI subsampling problem." is overly strong, considering that only one dataset and one solution method was used. Furthermore, the experiment regarding SNR merely shows a correlation between high gradient noise and underperformance in non-greedy methods, not necessarily a causal relation. Overall, I agree with the authors' rebuttal comment in lines 5-7 about what their primary claims are, but I do not think the paper is written in accordance with what they said in this rebuttal. The writing in introduction and conclusion are written with more definite statements than implied by the stated claim that "variance provides at least a **partial** explanation for the greedy/non-greedy performance gap". I suggest that the authors tone down the greedy vs. non-greedy discussion, and place greater emphasis on the more general policy gradient solution methodology as the main contribution of this work.


Review 3

Summary and Contributions: The paper proposes a novel approach for optimizing sampling in accelerated magnetic resonance imaging (MRI). They propose to leverage deep policy gradient methods, and show improvement over competitive baselines. In addition, they conjecture that the counter-intuitive result that greedy policy gradient approaches outperform non-greedy one originates from the presence of noise in the non-greedy objective's gradients. They empirically verify their claim by studying the signal-to-noise-ratio of the gradients of their greedy and non-greedy methods.

Strengths: The problem of optimizing sampling for MRI has become vastly investigated in the last years, and several exciting applications of experiment designs or reinforcement learning could find an application to MRI, which is very valuable to demonstrate their value in medical applications. It is also surprising that very few works have tried to apply RL methodologies to MRI sampling optimization, so this work could also serve as a proof-of-concept. The proposed approach of leveraging policy gradient methods is a natural approach to tackle the problem of sequentially optimizing sampling pattern with a discrete distribution of locations to be acquired. The results show small, yet consistent improvements over the baselines - including competitive ones - on different horizons, and the analysis of the noise in the gradients helps confirm the hypothesis of the authors.

Weaknesses: The main claim of the paper is the hypothesis that the noise in the non-greedy objective's paper is the reason why the greedy method can outperform it. However, I think that the empirical methodology is not strong enough to back this claim up, as the experiments are carried out on a single dataset, using a single network architecture and reported with a single performance metric. I think that the hypothesis would be much more clearly substantiated if the noise in the gradients were shown to be a consistent trend in various setting; I am afraid that, in the current state, the conclusion could be an anecdotical performance of the given setting. In addition, if I'm correct, RL models are prone to unstable training and are generally hard to train well. How can you confidently ensure that this behaviour isn't due to the RL policy not being trained for long enough? I also think that the experimental validation could be deeper. In particular, I think that the results with $\gamma \in (0,1)$ should be considered. It would be interesting to see whether a more greedy-like setup (gamma closer to 0) consistently shows less noise in the gradient than a non-greedy-like (gamma closer to 1) one, and this would strengthen the author's claim. Such questions should, in my opinion, be addressed in the paper. ----- AFTER REBUTTAL: I want to first thank the authors for their reply. After reading the reply and further discussing the paper with the reviewers, I will raise my score to weak accept. The other reviewers convinced me to reconsider the proof-of-concept of RL-based solutions for MRI optimization as a strong enough contribution on its own. Also, outperforming the MCTS-based approach is a strong result. I believe that the larger scale experiments and the exploration of gamma in [0,1] will make the paper stronger overall. However, I am still convinced that in its current form, the paper puts forth claims that are too strong regarding the noise in the gradient being the reason of the superior performance of greedy methods. I strongly encourage the authors to reword their statement (see for instance the abstract) to emphasize that what they provide is a possible, partial explanation of the phenomenon. I would expect the hypothesis to be validated on another dataset at the very least before claiming to have "experimentally verif[ied] that this noise leaves the non-greedy models unable to adapt their policies".

Correctness: As mentioned above, my biggest fear is that the result wouldn't hold as clearly in other settings.

Clarity: The paper is mostly well written, and the MRI-specific jargon is well defined and clear to understand. I have a couple of suggestions that could maybe improve the clarity of the presentation. - In the theoretical part, I think that some notation might be confusing for the reader. s and a are used to denote state and actions respectively, but S and A denote a subsampling operator and the reconstructed data, whereas in standard the likes of Sutton's RL book, this usually denotes the random variable to state and actions. Maybe using something else could disambiguate this? - In my opinion, equation 2 does not really capture the idea that you want your policy to give you the best prediction at each step to overall reach the best possible performance when you exhaust the sampling budget. - It is not totally clear how the oracle baseline works. It is said to be non-adaptive, but I don't understand how it is obtain? Do you greedily compute on the test set the location which will improve most your SSIM - From an experimental perspective, I think that appendix A does a really nice job at providing most needed information to reproduce the results. However, I would consider adding more details on the training itself. First, I think that you could provide slightly more details about the parameters that you used (e.g. parameters of Adam, restatement of the objective functions, ...). Secondly, I think that adding a section on the training of the RL models themselves would help: I know that it's common practice to use replay memories (maybe it's not applicable there) and things like that to make the training of the model more stable. An pseudo-code like description of the detailed training procedure would certainly be useful.

Relation to Prior Work: The discussion is overall clear. I noticed two small points that might be imprecise in related literature: - [30] is cited in the introduction and in section 2, but it seems that from the description around line 94, that the model uses a single network for conditional generation and prediction of the next move, which is not what is stated around line 42. - I think that [10, 11, 29] aim mainly at providing a greedy, non-adaptive sampling mask from scratch, and the main benefit that they bring is that they do not depend on a heuristic distribution, but rather build their estimate only through the training data. I also wonder why the authors did not connect the work more explicitly to the REINFORCE [1] approach and did not acknowledge the host of research carried out in trying to reduce the noise in the gradient variance. See for instance [2] and references therein. [1] Williams, Ronald J. "Simple statistical gradient-following algorithms for connectionist reinforcement learning." *Machine learning* 8.3-4 (1992): 229-256. [2] Kool, Wouter, Herke van Hoof, and Max Welling. "Estimating gradients for discrete random variables by sampling without replacement." *arXiv preprint arXiv:2002.06043* (2020).

Reproducibility: No

Additional Feedback: Remarks. I find it strange that this paper, which is about MRI sampling optimization, does not show any reconstructed image. I think that a visual assessment of the different methods could be useful. Questions. - Could you elaborate on why did you chose to use SSIM over PSNR? - In Table 2, do you have an idea why is the SNR largest for greedy at epoch 20? - As the paper is built around the noise in the different approaches, couldn't experimenting with different baselines in (5) and (6) actually help reduce the variance in the gradients? Possible Mistakes. - In equation (2), shouldn't m+1 be replaced with M? l. 136 speaks of improved quality after M steps. - l. 280, shouldn't the denominator be 1/(B-1) and not 1/B(B-1)? Notation. - r(A_{t'}, A_{t'+1}) does not explicitly show the dependence on x, maybe putting it as a subscript could make the dependence clearer?

[Author Response · NeurIPS 2020]

**Experimental verification (R2, R3).** Although we agree that additional experimental verification would be valuable, we believe our conclusions are warranted. Before submission, we performed initial experiments to test the stability of our conclusions and the RL method: more epochs with various learning rates, larger policy models, and adding subsampling location features to the policy model input. We observed nothing that changed our conclusions. We would characterise the primary claims of our paper as a proof-of-concept for policy gradient methods, as well as the insight that adaptivity is key, and that variance provides at least a partial explanation for the greedy/non-greedy performance gap. We agree that additional explanations for this gap may exist, but believe our primary claims as stated are sufficiently supported. We explored various other reward baselines: a running average baseline for each acquisition step, both per state and averaged over states, and a baseline for the non-greedy model that used full returns of parallel trajectories. These approaches all underperformed the reported baseline. We additionally explored an extension to our reported baseline that samples actions *without replacement* and computes gradients using the estimator in [22]. This performed on par with our reported method. We chose to train on SSIM reward because it typically corresponds to human evaluations of image quality more closely than PSNR [21]. Interestingly, evaluating our SSIM non-adaptive and adaptive oracles on PSNR gives scores of $27.21, 27.50$ on the base horizon, and $25.59, 26.13$ on the long horizon task respectively, whereas SSIM scores were clearly higher for the long horizon task, and indeed this is what one would expect for oracles. This suggests that SSIM and PSNR care about distinct features (notably, PSNR seems to favour more low-frequency columns), which complicates drawing conclusions from PSNR evaluations of SSIM optimised methods: verification of reconstruction quality by human experts may be necessary. To this end, we will include example reconstructions in the paper ready version. We furthermore plan to add experiments on the fastMRI brain data mentioned by **R2**. Additional experimental results for $\gamma \in [0, 1]$ would indeed be valuable to verify the observed SNR trend that underpins our conclusions on gradient variance, and we plan to include an analysis of this in the camera-ready version (**R3**). Finally, as per suggestion of **R1**, we will include an equispaced baseline in Table 1: initial results indeed show improvements over the random baseline, but our models still dominate in performance.

**Scaling to larger images (R2).** We ran initial experiments on larger $256 \times 256$ images, which indicated that our models are still able to learn performant policies. We used the raw k-space data for all our experiments, and chose to use the smaller image setting for our final experiments solely due to computational constraints.

**Differences with [18, 44] (R2).** The approach in [18] uses an RL based method in which the reconstruction and policy models can be decoupled. Unlike our approach however, MCTS based training does not naturally allow for training greedy models. This aspect is crucial to our further analysis, which indicates that greedy models may be favoured. Additionally, our approach enjoys a computational advantage due to the use of smaller models and converges more quickly. Finally, direct policy optimisation involves fewer design choices than computing an MCTS distribution. We will include pseudo-code of our training process that shows additional discrepancies, such as the omission of a replay buffer (**R3**). The approach in [44] requires joint training of the reconstruction network with an evaluator network that guides acquisition through a similarity score between ground truth and fantasised k-space. Joint training is crucial, as the reconstruction network must be incentivised to produce reconstructions that have correct k-space representation for evaluator based acquisition to perform well. This contrasts with our method, where joint training is optional, and our acquisition function is directly (reinforcement) learned using policy gradients on image-space input. This also poses a challenge for making a fair comparison (using the same reconstruction model): the reconstruction model in [44] is incentivised to care about features that are not necessarily relevant to our policy, and our reconstruction method is not necessarily incentivised to care about features that are crucial to their evaluator. We did a proxy comparison using our reconstruction model and replacing their evaluator score with the true spectral map score computed from ground truth images. Using ground truth test images makes this an oracle method - infeasible in practice - but provides an upper bound for the performance of [44] under our reconstruction model, as we now use true spectral map scores, rather than the estimate learned by the evaluator network. However, this oracle method performed far worse than our models, suggesting that the strategy in [44] indeed depends heavily on reconstruction model design choices that force consistency of k-space, as well as on joint training with the evaluator. We also note that there is no code available for [44], further complicating attempts at a fair comparison. We will include this discussion in our paper.

**Equation (2) (R2, R3).** Equation (2) indeed erroneously conflates $m$ and $M$. We will include the fixed formulation:

$$\pi^* = \arg\max_{\pi} \mathbb{E}_{\boldsymbol{x} \sim \mathcal{D}}\left[\eta(\boldsymbol{x}, A_\theta(S_M F \boldsymbol{x}))\right], \quad S_M = [k_1, k_2, ..., k_M]^\mathsf{T}, \quad k_m \sim \pi(\boldsymbol{y}_m).$$

Finally, we thank the reviewers for indicating where the paper could be clearer in notation and contains inconsistencies in the discussion of related works: these will be addressed. We will furthermore include the suggested references. To answer some final questions: **R2**: In equation (5) $\gamma$ is set to 1. A factor $\gamma^{t-t'}$ should indeed be included inside the sum over $t'$ in general: we will clear this up. **R2**: In the MDP formulation as presented the reward indeed depends on the ground truth image, and transitions are only deterministic when additionally conditioned on this. We - like the reviewer - do not expect this point to affect our conclusions, but will fix it in the final paper. **R3**: The denominator $\frac{1}{B(B-1)}$ follows from equation (13) in [4], where we are computing $\hat{\sigma}_{\hat{\mu}}^2 = \frac{1}{B}\text{Var}[\hat{g}]$, with $\text{Var}[\hat{g}] \approx \frac{1}{B-1}\sum_i (g_i - \hat{\mu})^2$.

[Meta-Review · NeurIPS 2020]

Three knowledgeable referees agree that the paper makes a valuable contribution to the MRI acceleration and reconstruction literature. They recognize the soundness of the proposed approach and its compelling experimental validation. I agree with the referees and recommend acceptance. However, please consider revising your paper to include R2's and R3's remark on toning down the claims related to greedy vs non-greedy, as well as R1's and R3's suggested references (https://arxiv.org/pdf/2007.10469.pdf, [1] Williams, Ronald J. and [2] Kool, Wouter, Herke van Hoof, and Max Welling).